# The Role of Copper Homeostasis in Brain Disease

**DOI:** 10.3390/ijms232213850

**Published:** 2022-11-10

**Authors:** Yumei An, Sunao Li, Xinqi Huang, Xueshi Chen, Haiyan Shan, Mingyang Zhang

**Affiliations:** 1Institute of Forensic Sciences, Suzhou Medical College, Soochow University, Suzhou 215000, China; 2Department of Obstetrics and Gynecology, The Affiliated Suzhou Hospital of Nanjing Medical University, Suzhou 215000, China

**Keywords:** copper, cuproptosis, brain injury, neurodegeneration, cognition

## Abstract

In the human body, copper is an important trace element and is a cofactor for several important enzymes involved in energy production, iron metabolism, neuropeptide activation, connective tissue synthesis, and neurotransmitter synthesis. Copper is also necessary for cellular processes, such as the regulation of intracellular signal transduction, catecholamine balance, myelination of neurons, and efficient synaptic transmission in the central nervous system. Copper is naturally present in some foods and is available as a dietary supplement. Only small amounts of copper are typically stored in the body and a large amount of copper is excreted through bile and urine. Given the critical role of copper in a breadth of cellular processes, local concentrations of copper and the cellular distribution of copper transporter proteins in the brain are important to maintain the steady state of the internal environment. The dysfunction of copper metabolism or regulatory pathways results in an imbalance in copper homeostasis in the brain, which can lead to a myriad of acute and chronic pathological effects on neurological function. It suggests a unique mechanism linking copper homeostasis and neuronal activation within the central nervous system. This article explores the relationship between impaired copper homeostasis and neuropathophysiological progress in brain diseases.

## 1. Introduction

In our brain, copper plays a key role in maintaining the redox balance of our most energetic organ [1]. Despite the high demand for metals in the brain, copper (Cu) is an important cofactor in electron transfer reactions and is an essential trace element for humans [2]. In addition, the accumulation of metal elements in tissues is associated with ageing or age-related diseases including cancers and neurodegenerative disorders (e.g., Alzheimer’s, Parkinson’s) and metabolic disorders (e.g., diabetes). Cu is required for many metabolic functions and it is crucial to find a way to regulate the metabolism of Cu in the human body. Cu is absorbed through the gastrointestinal tract, stored in the liver, and mobilized into the blood, but how Cu is maintained in the whole body is poorly understood [3]. As a critical element, Cu exists in two oxidation states, Cu^+^ and Cu^2+^. Cu has been exploited for its redox property during the evolution of Cu-containing enzymes. From mitochondrial oxidative phosphorylation to peptide hydroxylase, they use Cu as a cofactor during hydrolysis, electron transfer, and oxygen-harvesting reactions [4]. Its redox properties make copper both beneficial and toxic to cells [5]. In addition to supporting normal cell physiology, copper is one of the essential micronutrients for living organisms, the most common foods containing Cu are shellfish, meats, seeds, nuts, lentils, leafy green vegetables, and cocoa [6]. For optimal human health, Cu is involved in several fundamental processes including respiration, connective tissue formation, wound repair, macronutrient energy metabolism, catecholamine biosynthesis, and iron flux. Physiologically, copper also plays an essential role in human metabolism. Deficiencies in copper affect cardiovascular development, brain and liver function, lipid metabolism, inflammatory response, and resistance to chemotherapy. The intracellular distribution of copper in human cells is regulated by metabolic demands and changed according to the cell environment, and it is accessible in several cell compartments [7]. Even though Cu plays an important role in physiological processes, high levels of Cu can cause health problems and may be toxic. Less than 2–5% of the copper in the body is free and/or bound to amino acids or peptides [8]. This part of copper is officially known as free copper and may be harmful to the human body due to its oxidation [9]. Cu levels within cells and tissues need to be tightly regulated. Excess or deficiency of copper can lead to serious illness or death [10,11]. As copper has a powerful redox capacity, which can lead to oxidative stress and neurodegeneration within the brain [12,13,14,15,16,17]. Indeed, recent advances in the observation of Cu signalling and metabolism across multiple organ systems in both healthy and diseased states highlight its importance in mammalian biology. Here, we focus on copper as a canonical example of a metal signal pathway, providing a summary of the current understanding of copper signalling in neurobiology and future prospects for the field.

## 2. Copper Metabolism in the Brain

As a key component of neuronal development, maturation, and functions, Cu may enter the brain through the Cu transporter located at the brain barriers in a controlled manner. The blood–brain barrier (BBB) and blood–cerebrospinal fluid barrier (BCB) regulate copper homeostasis in the brain [18,19]. A major route for Cu to enter rat brain parenchyma was identified as the blood–brain barrier. Specifically, it was found that the blood–CSF barrier fine-tunes Cu homeostasis in the brain [20]. Approximately seventy percent of copper imported into mammalian brain cells is handled by CTR1 [21]. If there is an excess of copper, then the excess copper is released from the brain cells into the cerebrospinal fluid (CSF) and is taken up by the cells that make up the blood–cerebrospinal fluid barrier (BCB). The copper taken up by these cells is either stored by ATP7B for potential transport to the CSF or transported into the blood by ATP7A(Figure 1A) [22].

The copper proteome was defined using gene sequence data [23,24]. In eukaryotes, the size of the copper proteome is usually less than 1% of the total proteome of an organism. The occurrence of copper-binding proteins is relatively scarce when compared to that of zinc-binding proteins and of non-heme iron proteins [25]. In the brain cells, copper is taken up into cells by the copper transporter CTR1. Based on its kinetic accessibility, cellular copper can be divided into two categories, the typical stationary pool (i.e., copper bound to enzymes such as CCO or SOD1) and the unstable pool (i.e., copper bound to chaperones such as CCS) [26]. Unlike the former form of copper, the latter form is more bioavailable and is capable of participating in dynamic cell signalling pathways [27]. Copper is transferred to the copper protein through the copper-metal chaperone ligand exchange reaction [25].

After copper enters the cell, it binds to the cellular copper chaperone CCS and is then transferred to SOD1, where it inserts a disulfide bond. As part of the secretory pathway, Atox1 transports copper to copper-transporting ATPases. The Cu-ATPases accept copper from Atox1 and use the energy of ATP hydrolysis to transfer copper into the secretory pathway, where copper is incorporated into copper-dependent enzymes. Cu-ATPases are phosphorylated by kinase-mediated kinases and relocated to vesicles near either the basolateral (ATP7A) or apical (ATP7B) membranes in response to copper elevation. Upon fusion of vesicles, copper is exported. Metabolic factors that induce copper uptake also promote Cu-ATPase trafficking [28,29,30]. It is ATP7B that carries out copper transport in liver cells and ATP7A that carries out copper transport mainly in brain cells [25]. MT1/2 binds to more than one copper ion and can act as a reservoir for copper. In addition, glutathione (GSH) can also be directly or indirectly involved in regulating the cellular copper pool. In the mitochondria, a small copper ligand (CuL) supplies Cu^+^ to the IMS [25]. In the IMS, COX17, SCO1, and COX11 form two copper transport pathways for CuA and CuB, participating in the metallization of the mitochondrial CCO complex and embedded in the IM. Nuclear-encoded mitochondrial proteins, such as unfolded COX17, are imported across the OM via the TOM translocase and then captured in the IMS, following the introduction of disulfide bonds (SS) through the actions of Mia40. A sulfhydryl oxidase Erv1 generates a reactive disulfide on Mia40. COA6 and SCO2 assist in keeping the redox balance of SCO1, which in turn helps maintain its copper binding and transport to COX (Figure 1B) [25].

## 3. The Physiological and Pathological Role of Copper in the Brain

The brain contains approximately 9% of the body’s copper, the third-highest concentration of copper of any organ [12,31]. Brainstem neurons in a small area called the locus coeruleus (LC) are primarily responsible for producing the neurotransmitter norepinephrine (NE), which is the brain area with the highest concentration of copper [1]. Copper modulates rest-activity cycles through the LC [6]. Neural pathways originating in the LC send a wide variety of signals throughout the brain, playing a major role in regulating vertebrate arousal and wakefulness [32]. Copper is oxidized in a wide variety of oxidation states, but the oxidation states Cu^+^ and Cu^2+^ are most common within cells, and Cu^2+^ is more common outside cells [12]. The distribution of copper in the brain is uneven, not only in the locus coeruleus region but it has also been recorded at higher levels in the substantia nigra [33]. A delicate homeostasis of copper in the central environment is maintained by the blood–brain (BBB) and cerebrospinal fluid barriers (BCB) [19]. According to histochemistry studies conducted on brain slices, glial cells have a higher copper concentration than neurons, both under physiological and pathological conditions [22,33,34]. Senescent cells accumulate intracellular copper irrespective of the source of stimulation or the origin of the cell, and this is likely a universal phenomenon [35]. A number of neurological disorders, including Wilson’s disease and Alzheimer’s disease, alter both the total copper level and the distribution of copper in the brain [22,36,37]. Therefore, copper plays an important role in the brain.

### 3.1. Copper and Inflammation

Since the brain has a high metabolism and signalling needs, copper is particularly abundant in this organ [12,38,39,40,41]. Endogenous copper plays an essential role in regulating inflammation [42]. Copper concentrations and CP activities in bodily fluids and tissues tended to rise in humans and animals under acute and chronic conditions of inflammation [43]. ATP7B transports Cu into CP before it is released into the plasma. CP is an acute phase response protein whose synthesis and secretion can be distinctly increased during inflammation. Hepatic synthesis of CP can be upregulated by inflammatory cytokines such as interleukin-1 (IL-1) and interleukin-6 (IL-6), as well as a hypoxia-inducible factor (HIF1) [44]. Furthermore, copper-deficient rats were more susceptible to the standard acute inflammatory agents than rats receiving a normal copper diet [45,46]. On the contrary, excess Cu contributes to numerous inflammatory vascular diseases, while Cu chelators inhibit inflammation [47,48]. Previous studies have worked on the inflammatory effects of copper in the liver, but few have focused on the effects of copper in the brain. Hitherto, three members of the MAP kinase family, such as p38 mitogen-activated protein kinase (p38 MAPK), extracellular signal-regulated protein kinase (ERK) and c-Jun N-terminal kinase (JNK), have been reported in mammalian cells. These kinases can facilitate the generation of pro-inflammatory cytokines [49]. Copper influx promotes MEK1 phosphorylation of ERK1/2. Activation of MAPKs triggers stimulation of other kinase targets, which are then translocated to the nucleus to activate the transcription of pro-inflammatory genes [50]. Inhibition of p38 and JNK helps to remove excess copper from hepatocytes [51]. Nuclear factor kappa B (NF-κB), a critical activator of inflammatory processes, could modulate the expression of various inflammatory mediators (such as interleukin-8, inducible nitric oxide synthase, interleukin-1β, cyclooxygenase-2) in numerous cells [52,53,54,55]. Therefore, both MAPK and NF-κB are essential elements of the pathways that regulate the inflammatory response. The imbalanced production of mediators with anti-inflammatory and pro-inflammatory functions also triggers the inflammatory response in the brain. Pro-inflammatory cytokines are essential for the development and maintenance of inflammation [54], which can lead to damage in multiple organs, including the brain [56,57,58]. On the other hand, TGF-β acts as an anti-inflammatory cytokine and has a suppressive effect on the inflammatory response [58]. It has been demonstrated that tissue copper levels are significantly increased under pathological inflammatory conditions [59,60]. A recent study confirms ATP7A as a therapeutic target for inflammatory vascular disease [61]. These copper-transporting ATPases (ATP7A/B) are responsible for maintaining intracellular copper levels. Intracellular copper influences the activities, post-translational modifications, and localization of copper-dependent proteins [62]. At the same time, there is growing evidence that most brain disorders show an inflammatory component [58,63,64]. Under inflammatory stimuli, a higher level of labile copper in microglia was observed [65,66,67]. In the past few years, it has been found that an anti-inflammatory effect of copper delivery in the chronic neuroinflammatory environment of a rat model of Alzheimer’s disease [68]. Furthermore, copper metabolism is significantly enhanced in the acute phase of inflammation [44].

### 3.2. Copper and Immunity

Copper performs a variety of functions in the immune system, among which the direct action mechanism of copper on the immune system is less known. In order to understand the role of copper in the immune response, a number of animal models and cultured cells have been used in experiments [69]. In general, the effectiveness of the acquired response will be reduced when there is a copper deficiency [70,71,72]. Early studies of copper deficiency identified that copper-deficient animals were anaemic, had markedly lower thymus weights, and markedly higher spleen weights than control animals [71]. Furthermore, the production of antibodies by spleen cells was markedly reduced in copper-deficient animals [71,72]. Lukasewycz and Prohaska found a significant increase in IL-1 and a significant decrease in IL-2 in copper-deficient rats [71]. In addition, mitogen-induced DNA synthesis is damaged by copper deficiency, which results from a reduction in IL-2 concentrations [73]. What is more, neutropenia has been known to be a sign of copper deficiency since the 1960s [74]. Higuchi et al. measured anti-neutrophil antibodies in the serum of copper-deficient patients, which might suggest a mechanism of neutrophil loss [75]. In addition to a reduction in the number of circulating neutrophils, the function of these neutrophils is also harmed in copper deficiency [76,77]. Furthermore, Cu/Zn SOD has been found in human neutrophils and monocytes [69]. Although it has been confirmed that copper deficiency has a negative effect on the human immune system, the specific mechanism needs to be further studied.

### 3.3. Copper and Oxidative Stress

Oxidative stress is caused by an imbalance between the production of reactive oxygen species (ROS) and antioxidant defences. Particularly, the brain becomes damaged with age and shows pathological changes in oxidant production or antioxidant levels in mammals [78]. On the one hand, several components of the oxidant defence system such as superoxide dismutase (SOD), CP, GSH, and metallothionein are impaired in copper deficiency. In addition, Cu/Zn SOD and CP activity are sensitive to tissue copper, as they require copper as a catalytic cofactor [79]. It has been reported that Cu deficiency causes a decrease in Cu/Zn SOD activity [80,81,82,83], but protein levels of Cu/Zn SOD may or may not be reduced [84,85,86]. Furthermore, it has been demonstrated that a 50–60% reduction in Cu/Zn SOD activity can lead to severe oxidative stress and cell death [87,88]. Under normal physiological conditions, the antioxidant defence system must be steadily regulated. CP is synthesised in the liver and copper is needed for the function of its ferredoxin enzyme [89]. Although the state of copper does not affect the synthesis or secretion of CP, its absence reduces its activity as copper cannot be incorporated into CP, making it less stable [89]. Low CP activity is a common feature of copper-deficient animals [90]. Furthermore, GSH is frequently increased in the liver and plasma of Cu-deficient animals [81,91], a change which is considered to indicate an adaptive response to increased oxidative stress. Metallothionein is also involved in the homeostatic control of copper, which can bind Cu and render it redox-active under reducing conditions [92].

On the other hand, under copper exposure, the antioxidant defences of fish fail because of the over-production of reactive oxygen species (ROS) [93]. The toxicity of excessive copper is mainly related to the production of ROS [94]. Nevertheless, the brain is rich in polyunsaturated fatty acids, which are especially vulnerable to attack by ROS [95]. Furthermore, copper-induced ROS consist mostly of superoxide and hydroxyl radicals [96]. Copper exposure increased ROS production, resulting in oxidative damage and reduced the fish brain’s ability to scavenge hydroxyl radicals. Copper exposure reduces brain GSH levels [93]. GSH could directly scavenge singlet oxygen and hydroxyl radicals to intact cells under oxidative stress [97]. The decrease in GSH can be attributed to two factors. Firstly, it might be partly owing to the increase in ROS production caused by Cu stress, which consumes a large amount of GSH [93]. It has been reported that in human gingival epithelial cells, excess ROS can deplete GSH, leading to a decrease in GSH levels [98]. Secondly, the decrease in GSH content may be partly because of the inhibition of GSH re-production [99]. What is more, Cu exposure increased the nuclear accumulation of Nrf2 in the fish brain and increased its ability to bind to ARE(Cu/Zn SOD). Furthermore, Cu exposure resulted in increased expression of Nrf2, MafG1 and PKCd genes, indicating that the de novo synthesis of these factors is necessary for the long-term induction of such antioxidant genes [93].

### 3.4. Copper and Cell Death (Cuproptosis)

Copper is involved in cell growth/proliferation and autophagy pathways. When Wilson’s disease or an abnormal buildup of copper in the liver occurs, copper inhibited cAMP degradation by directly binding to a conserved cysteine residue in the phosphodiesterase3B (PDE3B), which breaks down triglycerides into fatty acids and glycerol [5]. Copper-dependent kinase signalling can regulate autophagy through ULK1/2. Copper metalloallostery promotes protein degradation and induces cell death through metalloallostery activation of the E2 binding enzyme UBE2D1-UBE2D4 [100]. It is reported that copper acts on MEK1/2 and enhances the ability of MEK1/2 to phosphorylate ERK1/2 [101].

Recently, a new type of cell death has been proposed: cuproptosis. The classification of cuproptosis was proposed by Tsvetkov et al. [102]. Cuproptosis and ferroptosis are characterized by distinct alterations in energy metabolism and mitochondrial function [103]. Cuproptosis is a novel phenomenon, which is a novel cell death pathway mediated by lipoylated TCA cycle proteins [104]. Cu ionophores, elesclomol (ES), could bind copper. It has been thought that elesclomol-induced cell death is mediated by an increase in mitochondrial ROS [55,105]. However, now it is thought that elesclomol binds to copper, enters the cytoplasm, and copper is reduced to univalent copper. Fe–S clusters are formed in mitochondria by FDX1, a mitochondrial reductase [106,107,108], a process that is essential for mitochondrial function [109]. One study used a genome-wide CRISPR-Cas9 screen to determine which gene loss makes elesclomol analogues resistant. Interestingly, two screenings yielded only one gene, FDX1. FDX1 encodes ferredoxin 1, whose underlying mRNA expression is highly correlated with elesclomol sensitivity [110]. Elesclomol is specific for FDX1, binding to the FDX1 α2/α3 and β5 chains, but not to its homologue FDX2 [110]. Fe–S proteins in the mitochondrial respiration chain deliver the electrons generated from the tricarboxylic acid cycle (TCA cycle) to ADP molecules for energy production, thereby complying with the high demand for energy consumption of neuronal cells [19]. Copper binds directly to the lipoylated components of the TCA cycle. The aggregation of these copper-bound lipoacylated mitochondrial proteins and subsequent loss of Fe-S cluster proteins then triggered cuproptosis [103]. Finally, in the above process, as a high-affinity Cu importer, CTR1 plays an important role in transferring copper into the cell (Figure 2).

## 4. The Copper Signal Pathway in Brain Diseases

### 4.1. Alzheimer’s Disease (AD)

Alzheimer’s disease is very common in older people [111]. Misalignment and imbalance of metal ions can lead to protein aggregation and reduced activity, and induce oxidative stress. There are various pathogenic factors that may cause AD by causing its development and progression [112]. The pathological features and clinical diagnostic criteria for AD are neuritic plaques and neurofibrillary tangles in excess of those found in age-matched healthy individuals [113,114]. Neuritic plaques are composed of a central core of amyloid protein surrounded by astrocytes, microglia, and dystrophic neurites often containing paired helical filaments. Neurofibrillary tangles are paired helical filaments containing abnormally phosphorylated tau proteins that occupy the cell body and extend into the dendrites [115]. However, amyloid β (Aβ) protein, which is thought to be central to the pathogenesis of AD, is derived from AβPP and is deposited in neuronal plaques [116]. AβPP is important for regulating Aβ production, while Aβ aggregates can produce ROS in the presence of copper ions and excess ROS is harmful to the brain. However, oral administration of copper chelator, Temozolomide (TM), significantly improved the cognitive decline of AβPP/PS1 Tg (transgenosis) mice and revealed that copper chelators promote the expression of ADAM10 and the production of sAβPPα via MT1/2 and its downstream Gq/PLC/PKC/ERK, Gs/cAMP/PKA/ERK and Gs/cAMP/PKA/CREB signalling pathways [117]. As well as the aggregation of proteins, dyshomeostasis of copper ions has been reported within AD brains. Copper concentrations in AD patients’ brains have been shown to be 400 mM. As a comparison, healthy brain tissues of the same age contained copper about 70 mM [118,119]. When Cu^2+^ binds to Aβ, it also produces reactive oxygen species, which leads to neuronal damage [120]. However, chelating copper by microglia may contribute to AD neuroprotection [65]. Accordingly, copper homeostasis can serve as a therapeutic target to prevent AD [117].

### 4.2. Menkes Disease (MD)

Menkes disease is caused by various mutations in ATP7A, a type1 ATPase that transports copper [121,122]. Cu^+^ is transported from the cytosol into the secretory pathway, or into vesicles, by ATP7A (also known as Menkes protein, MNK) [121,122,123]. Cu^+^ is then incorporated into lysyl oxidase and tyrosinase, both of which are copper-dependent enzymes. Copper-dependent enzymes are deficient, which causes many symptoms of the disease [124]. The Cu^+^ pumping into the vesicles in the latter case is then released into the extracellular environment after the membranes of the vesicles fuse with one another. In order for both roles to play correctly, the intracellular localization of ATP7A has to be controlled, which is influenced by the concentration of Cu^+^ within the cell [125]. Designed to transport copper to mitochondria, elesclomol increases cytochrome c oxidase levels in the brain. The action of elesclomol prevents neurodegeneration and improves survival in a murine model of severe Menkes disease (mottled-brindled mouse) [126]. There is well-documented evidence that brain copper depletion occurs in MD patients and mice models of this disorder. Copper levels in MoBr/Ybrain are decreased by 2- to 4-fold with age [127,128]. Copper-deficient 4-week-old murine brains showed no change in SOD1 levels [129]. However, the ATPA7A mutant mouse exhibited an increase in SOD3 levels in its aortas while SOD1 levels remained unchanged [130]. Low levels of ATP7A transcription were found in Purkinje cells of hippocampal and pyramidal neurons of the midbrain, which are most susceptible to neurodegeneration [131]. Immunoblot analysis of homogenates of wild-type and MoBr/y brains revealed only minor differences in total amounts of ATP7A protein, immunofluorescence revealed significant differences between cell types expressing ATP7A [132]. An interesting copper histochemical stain uses silver sulfide and trichloroacetic acid combinations for detecting copper in the macular and MoBr/y brains, neuronal populations deficient in ATP7A had a decreased amount of copper [133,134]. On the other hand, mutant ATP7A levels are dramatically higher in MoBr/y brain capillaries. A greater concentration of copper is found in cerebral endothelial cells in the macular and MoBr/y brain [133,134]. As ATP7A levels correlate with copper levels in diverse cell types, copper may also regulate ATP7A gene expression [135].

### 4.3. Wilson’s Disease (WD)

As a typical disorder of copper metabolism, increased copper levels have been demonstrated in the brain and liver of patients with Wilson’s disease [136]. WD is a genetic disorder that affects copper metabolism. There are approximately 1:7000 to 30,000 live births diagnosed with WD, making it one of the most common inherited liver disorders [137]. WD is caused by mutations in ATP7B, which encodes the transmembrane copper-transporting ATPase 2 (widely known as ATP7B), which mediates the excretion of copper into bile and provides copper for the synthesis of CP [138]. Wilson disease happens when there is an abnormal accumulation of copper in the body caused by hepatic failure to remove it. Excess Cu in this disease causes brain damage [11,139]. In recent years, researchers have exploited the relationship between CCC2 and Fet3 in the study of the Menkes protein; the expression of Menkes proteins in cells lacking CCC2 [140]. Associated with copper overload, WD primarily affects the liver and brain, although it may also manifest in other organs such as the cornea and kidneys, although to a lesser degree. Hepatocytes fail to excrete copper into the bile as a result. The disease is caused by a mutation in the ATPase 7b gene on chromosome 13q14.3 [141]. Cu accumulation is highly toxic since it is capable of damaging various intracellular components and disturbing cellular redox balance [142,143]. Cu can also damage mitochondria, a defect that occurs often in Wilson disease, as abnormal mitochondria disrupt the synthesis of metabolites that regulate epigenetic expression [144,145]. The liver and brain are most affected by copper toxicity when copper cannot be excreted from the body [146].

### 4.4. Traumatic Brain Injury

Traumatic brain injury (TBI) has increasingly become a major cause of morbidity and mortality worldwide, mainly occurring in traffic accidents, wars, or violent collisions among people [147,148]. Copper is essential for wound repair and regeneration, and higher-than-normal levels of copper have been detected in wound tissue [149,150]. Peng et al. explored increased copper uptake as a biomarker for the noninvasive evaluation of traumatic brain injury disease (TBI), and ^64^Cu uptake in the injured cortex was assessed with ^64^CuCl_2_ PET/CT. The results showed that the content of cortical copper in the TBI-injured group was significantly higher than that in the uninjured group [151]. Therefore, increased copper in injury brain tissue may be a new marker for assessing TBI. After TBI, Cu/Zn SOD also increased significantly [152]. SOD, as an endogenous free radical eliminator, can reduce brain injury after ischemia and TBI [153]. Shigeki Mikawa et al. demonstrated the neuroprotective effects of Cu/Zn SOD on cortical contusion in mice through transgenic mice, including acute injury, such as BBB destruction and brain oedema, and chronic injury, including functional motor recovery and tissue necrosis [152]. The occurrence and development of brain oedema after TBI is closely related to superoxide anion, and exogenous lecithin superoxide dismutase can clear superoxide anion, thus reducing the degree of brain injury [154]. Mitochondrial dysfunction induced by superoxide anion radicals contributes to the formation of damage in the mouse brain after physical trauma [155]. Serum ceruloplasmin and copper may be early markers of elevated intracranial pressure after traumatic brain injury [156]. Copper deficiency in the diet of rats and mice significantly impairs the central nervous system’s ability to cope with injury [157]. Maintaining copper homeostasis in the brain can be used as a target for the treatment of TBI [158]. The results of copper homeostasis imbalance after TBI are shown in Figure 3.

### 4.5. Intracerebral Hemorrhage (ICH)

Copper may play an important role in ICH. Decreased serum CP and increased serum free copper are associated with death or poor prognosis in hypertensive ICH patients [159]. Apoptosis or cell death after transient focal cerebral ischemia may involve ERK1/2 phosphorylation and SOD1 may be involved in attenuating mitogen-activated protein kinase/ERK pathway mediated apoptotic cell death [160]. Copper has angiogenic potential to promote skin wound healing [161]. The increased content of free radicals and reactive oxygen species plays a crucial role in ICH injury [162]. Takuma Wakai et al. have demonstrated that SOD1 overexpression plays an important role in neural stem cell survival after ICH brain transplantation. It is suggested that endowing neural stem cells with antioxidant properties is a possible way to improve the efficacy of ICH cell transplantations [163]. There is sufficient evidence to suggest that iron release from haematoma following glutamate release from erythrocytes and inflammatory response are major factors in ICH-induced brain injury [164]. However, iron and copper are closely related to the need for polycopper ferrous oxide [165].

### 4.6. Ischemic Stroke

Stroke is the second leading global cause of death behind cardiovascular disease (CVD) [166]. Lai et al. found small molecular copper and its related metabolites in the serum of patients with ischemic stroke [167]. Plasma copper and other metals were found to be associated with a higher risk of ischemic stroke in the study [168]. Furthermore, long-term exposure to water containing trace amounts of copper increased ischemic damage in mice, possibly in part due to damage to endothelial progenitor cells and a reduction in ischemic cerebrovascular production. So copper contamination in drinking water may be a risk factor for stroke [169]. Yang et al. found that the risk of stroke decreased with increased dietary copper intake [170]. In hypertensive patients in China, baseline plasma copper was positively associated with the risk of the first stroke, especially in some patients with higher BMI [171]. Hu et al. emphasized the need for research to determine the optimal range of plasma copper concentrations in Chinese people, as it may provide more specific clinical and nutritional guidelines for optimal copper levels for stroke prevention [172]. Abnormal Cu/Zn and Cu/Se molar ratios can be used as important indicators of nutritional status and oxidative stress levels in patients with acute ischemic stroke [173]. The copper complex CuII(atsm) possesses neuroprotective properties, as demonstrated in vitro, halting excitotoxic damage and protecting the N2a cells from oxygen and glucose deprivation, to be protective against permanent and transient ischaemia models in mice. Ischemic brains delivered with copper exhibit suppression of inflammation, specifically affecting myeloid cells. A reduction in CD45 and Iba1 immunoreactivity as well as changes in the morphology of Iba1 positive cells are observed in ischemic brain tissue. In addition, CuII(atsm) decreases invading monocytes by protecting endogenous microglia from ischemic insults and protecting endogenous microglia from ischemic insults. The results indicate that CuII(atsm), a copper complex, is an inflammation-modulating compound with high therapeutic potential for stroke and is a strong candidate for the development of treatments for acute brain injury [68].

### 4.7. Spinal Cord Injury (SCI)

Spinal cord injury leads to severed axons and neuronal death, resulting in permanent functional impairment [174]. SCI leads to massive cell death and damage to the blood–spinal cord barrier, then the infiltration of immune cells. Inflammation, the formation of free radicals leads to secondary damage, killing other cells such as oligodendrocytes [175]. After SCI, endonuclease G and apoptosis-inducing factors are transferred from mitochondria to the nucleus. Overexpression of SOD1 in transgenic rats can increase SOD activity in mitochondria and promote the survival of motor neurons after SCI by decreasing the release of endonuclease G [176]. Many researchers have found abnormalities in mitochondrial morphology and function in the spinal cord of patients with motor neuron disease [177,178,179,180,181,182,183,184,185]. It appears that SOD activity increases in the brain, reducing the development of vasogenic brain oedema and infarction [186]. When rats were injured, the amount of mRNA for CP increased significantly [187]. This copper-containing enzyme is widely found in numerous types of eukaryotes, containing six copper atoms [188]. CP can remove ROS through the activity of oxidase or peroxidase enzymes such as ferric oxidase, cuprous oxidase, and glutathione peroxidase [189,190,191]. The results of Wu et al. showed that CP expression was significantly increased in GFAP^+^ astrocytes, CD11b^+^ microglia, CNPase^+^ oligodendrocytes, NeuN^+^ neurons, CD45^+^ leukocytes and CD68^+^ activated microglia/macrophages after SCI. Quantitative analysis showed that neurons and oligodendrocytes did not participate in the CP elevation induced by SCI. However, the main sources of CP elevation are infiltrating leukocytes, activated microglia/macrophages, and astrocytes [192]. Inflammatory, traumatizing, or infectious conditions induce the induction of CP as a positive acute phase protein [193]. Studies on Cu’s likely physiological role in TSCI are scarce, but their role in protecting neurological tissue appears to be critical [193]. Moreover, premature death from congenital defects of Cu transporters (ATP7A) in Menkes disease and progressive neurodegeneration due to CP deficiency cause fatal neurological consequences, this study provides further evidence for the importance and impact of Cu-dependent proteins on neuron survival [194]. CP-deficient mice have been reported to have significantly increased motor neuron loss and show impaired primary motor recovery after injury [195]. Furthermore, they concluded that there was a strong association between temporal changes in copper status and clinical outcomes after traumatic spinal cord injury [193].

### 4.8. Glioma

Copper is an essential cofactor in angiogenesis and has been experimentally targeted for glioblastoma [196]. Human Cu/Zn SOD cDNA was transfected into U118-9 human malignant glioma cells. Compared with the wild-type and vector control, the Cu/Zn SOD activity levels of the four supertransfected cell lines were increased by 1.5, 2.0, 2.6 and 3.5 times, respectively. It is confirmed that Cu-Zn superoxide dismutase is a novel tumour suppressor gene [197]. In preclinical experimental proof-of-principle studies, copper reduction inhibited malignant tumour growth and invasion within the brain by inhibiting angiogenesis [198,199]. Endothelial cells proliferate when copper ions are present [200], and copper contributes to angiogenesis in tissues [201]. Basically, elesclomol functions by redoxing copper ions [202]. Reduced copper inhibits the actions of structurally diverse angiogenic factors, cytokines, and prostaglandins [201,203]. An in-depth investigation of the molecular mechanisms underlying glioblastoma stem-like cells (GSCs) and GSC-derived endothelial cells (GdECs) response to elesclomol found that this compound induces a strong increase in ROS in both GSCs and GdECs leading to non-apoptotic copper-dependent cell death [202]. Elesclomol acts on cancer cells by causing them to become apoptotic through the production of ROS [105]. In biological experiments, elesclomol is hypothesized to generate ROS by chelating copper and preventing redox cycling of copper [204]. Elisclomol binds to Cu^2+^, delivering it into mitochondria. From there it is converted to Cu^+^, which can bond with oxygen to produce ROS, A high level of free radical production leads to uncontrolled oxidative stress and apoptosis in cancer cells [205]. As evidenced by experiments on human keratinocytes [105] and in PBMCs [204], elesclomol is more effective against tumour cells with high ROS levels than against melanomas, but it is not toxic to normal cells. In many instances, tumour cells produce more ROS than normal cells [206,207]. Studies have shown that oxidative stress is the primary mechanism of elesclomol’s action on stem-like cells of glioblastomas and on endothelial cells derived from those stem-like cells. Ecoli cells treated with eclorophenol showed altered mitochondrial membranes, higher ROS production and a decrease in GSH levels [202]. Disulfiram can be used as an anticancer drug and a radiosensitizer [208]. Earlier studies have shown that disulfiram induces cytotoxicity via oxidative stress [209,210], which may be enhanced by the presence of copper [209]. The antiprotease effects of copper-binding drugs have been demonstrated [211], along with the formation of ROS [212]. Copper is chelated by disulfiram, so the copper–disulfiram complex may be toxic [213]. Copper is present in many tumours [211], and its role in tumour cells is also significant.

### 4.9. Other Diseases

Copper is essential for diverse neuronal functions. Copper induces microglia activation in substantia nigra pars compacta of C57BL/6J mice. In addition to this, copper activates BV2 cells and induces the release of inflammatory cytokines. In BV2 cells, copper induced oxidative stress and activated the NF-κB/P65 pathway, which interfered with mitochondrial autophagy and eventually led to BV2 cell death [214]. The most abundant glial cells in the central nervous system are astrocytes, which play important roles in health and disease. Under normal conditions, astrocytes are involved in important physiological processes, such as the development and functional regulation of synapses and the blood–brain barrier, metabolic support of neurons, and production of neurotrophins [215,216]. Kardos et al. believe that cell-level copper signalling between neurons and astrocytes is also present and may play an important role in brain signal processing [217]. Considerable evidence has shown that memory deficits in rats with chronic copper poisoning are associated with copper deposition in the choroid plexus, astrocyte swelling, astrogliosis and neuronal degeneration in the cerebral cortex, and increased copper levels in the hippocampus [218]. Particularly, in some neuroinflammatory diseases, such as multiple sclerosis, the expression of copper transporters such as CTR1 on glial cells depends on TRKB, and TRKB has been shown to play a key role in neurotrophin-induced calcium flux production in glial cells and CTR1 upregulation in vitro. These processes cause astrocytes to take up and release copper, which in turn leads to oligodendrocyte loss [219].

## 5. The Drugs for Copper

Copper-containing drugs work in two ways: by supplementing copper, or by chelating it. The application of copper in the laboratory is still limited, and experiments show that copper has a unique role. Clinical trials have been conducted in many areas related to brain diseases, but the results have not been as satisfactory as expected (Table 1).

### 5.1. The Increase of Copper

Copper is required for cell survival and proliferation and plays a very important role in the development and progression of brain diseases. In some copper-deficient diseases, appropriate copper supplementation is beneficial for injury recovery. Several pharmacological agents that supplement copper have been listed (Table 1).

It has been reported that copper deficiency in mammals causes serious impairment of cognitive and motor function [220]. In Menkes disease, using hydrophilic compounds to restore normal Cu levels and enzyme functions through parenteral Cu supplementation, copper histidine (HIS-Cu^2+^) is one example [221,222]. In addition, the copper transporter gene, ATP7A, is affected by a variety of mutations. Copper injections could prevent death and illness when administered early (ClinicalTrials.gov number, NCT00001262) [221]. DPy is a copper carrier that binds to and carries copper ions into cells and can act as a recyclable copper carrier, promoting intracellular copper accumulation and causing oxidative stress-mediated apoptosis in cancer cells [223]. In addition, despite the fact that KRAS raises intracellular Cu levels, the mechanisms behind this remain unknown [224]. A depolarized neuron releases copper into the synaptic cleft, which leads to local concentrations of up to 250 mM [225]. Cu deficiency has been linked to severe neurological deficits, and premature death, regardless of whether the cause is genetic or nutritional [6]. Copper deficiency can affect the body’s immune function, cause inflammatory disease or cause oxidative stress, which can lead to brain disease. Therefore, copper supplementation may improve diseases caused by copper deficiency.

### 5.2. The Decrease in Copper

High concentrations of copper are harmful because they promote a Fenton-like reaction. This leads to oxidative damage to all cellular components, proteins, lipids, and nucleic acids [7]. In AD, an excellent treatment is the use of a chelating agent that selectively removes Cu from Cu-Aβ [226]. Treating a mouse model of Wilson’s disease with DPM-1001 reduced copper levels in the liver and brain, removed excess copper through faecal excretion, and improved symptoms associated with the disease [227]. As there have been few in vivo studies of metal chelators, it is not yet possible to know with certainty what specific effects they have on disease progression [228].

## 6. Conclusions

This review elucidates the role of copper in inflammation, immunity, oxidative stress, and copper poisoning. This further illustrates the relationship between copper and brain diseases through the above processes. In addition, some common copper drugs are also discussed. However, the exact mechanism of copper-induced brain disease needs further investigation.

## Figures and Tables

**Figure 1 ijms-23-13850-f001:**
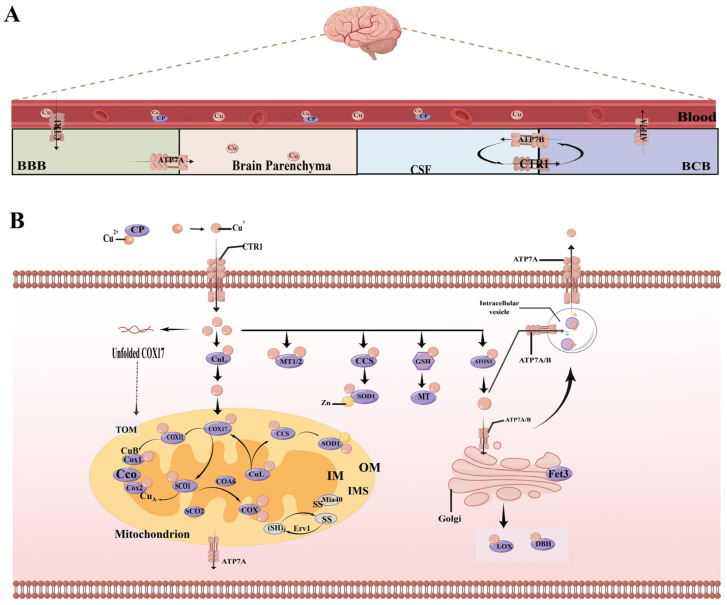
(**A**) The entry and exit of copper in the brain. Copper enters the brain through the blood–brain barrier (BBB). The endothelial cells that make up the BBB get copper from the blood via the apical copper transporter1 (CTR1) and transport it to the brain parenchyma via ATP7A. If there is an excess of copper, then the excess copper is released from the brain cells into the cerebrospinal fluid (CSF) and is taken up by the cells that make up the blood–cerebrospinal fluid barrier (BCB). The copper taken up by these cells is either stored by ATP7B for potential transport to the CSF or transported into the blood by ATP7A. (**B**) Copper metabolism of brain cells. Ceruloplasmin (CP) carries the copper to its destination. On the plasma membrane, copper ion channel CTR1 can achieve a high affinity for copper uptake. After copper enters the cell, a small copper ligand (CuL) supplies Cu^+^ to the mitochondria intermembrane space (IMS). In the mitochondria, copper chaperone for cytochrome C oxidase 17 (COX17) supplies two pathways, delivering copper to COX11 and synthesis of cytochrome oxidase1 (SCO1). Copper reaches the CuB site of the COX1 subunit via COX11 and the CuA site of COX2 via SCO1, participating in the metallization of the mitochondrial cytochrome C oxidase (CCO) complex and embedded in the inner membrane (IM). Nuclear encoded mitochondrial proteins, unfolded COX17, are imported across the outer membrane (OM) unfolded via the TOM translocase and then captured in the inner membrane space (IMS), following the introduction of disulfide bonds (SS) through the actions of Mia40. A sulfhydryl oxidase Erv1 generates a reactive disulfide on Mia40. Copper chaperone for cytochrome c oxidase (COX) is catalyzed by SCO1 and SCO2 which are metallochaperones. Cytochrome c oxidase assembly factor 6 (COA6) and SCO2 assist in keeping the redox balance of SCO1, which in turn helps maintain its copper binding and transport to COX. In the cytoplasm, metallothionein 1/2 (MT1/2) binds to more than one copper ion and can act as a reservoir for copper. Copper chaperone for superoxide dismutase (CCS) delivers copper to Cu/Zn superoxide dismutase (SOD). In addition, glutathione (GSH) can also be directly or indirectly involved in regulating the cellular copper pool. Copper ions bind to antioxidant protein 1 (Atox1), which presents copper to the ATP-driven transmembrane copper ion pumps ATP7A and ATP7B, both of which perform both copper export and metallochaperone functions, with ATP7B performing copper export in hepatocytes and ATP7A primarily performing copper export in brain cells. Together these proteins maintain proper intracellular copper bioavailability and ensure the metalation of copper-dependent enzymes including COX, superoxide dismutase 1 (SOD1) and oxygenases/oxidases including tyrosinase, lysine oxidase (LOX), dopamine β-hydroxylase (DBH). The figures in this article are all drawn by Figdraw.

**Figure 2 ijms-23-13850-f002:**
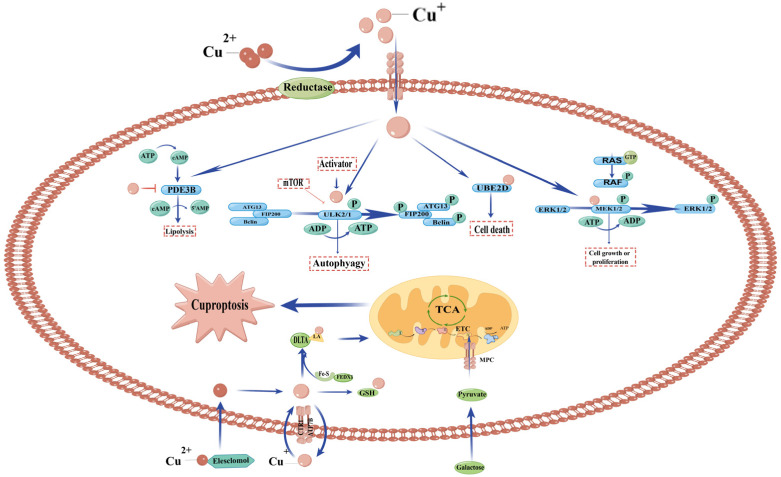
Copper participates in cell death and proliferation pathways. Copper binds to and inhibits Phosphodiesterase 3b (PDE3B), inhibits cyclic AMP (cAMP) degradation, and promotes cAMP-dependent lipolysis, which is needed for fat metabolism. Copper-dependent kinase signalling can regulate autophagy through ULK1 and ULK2. The copper signal promotes protein degradation by binding the E2-binding enzyme UBE2D1-UBE2D4. Copper-dependent kinase signalling can regulate cell growth/proliferation through MEK1 and MEK2. Furthermore copper binds directly to the lipoylated components of the TCA cycle. The accumulation of these copper-bound lipoacylated mitochondrial proteins and the following loss of Fe-S cluster proteins then triggered cuproptosis.

**Figure 3 ijms-23-13850-f003:**
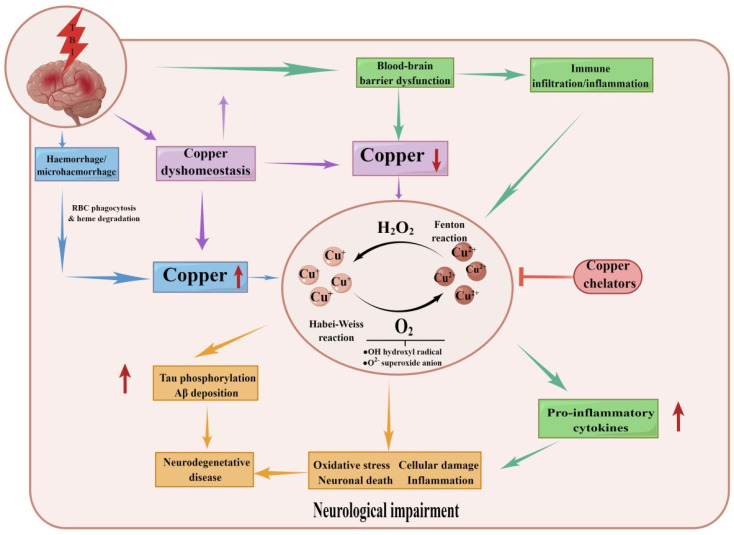
The outcomes of imbalanced copper balance after TBI. TBI leads to several serious consequences, including BBB breakdown, haemorrhage, and copper dyshomeostasis. Together this leads to a copper increase or decrease in the brain. Copper is involved in the Haber–Weiss/Fenton reaction, promoting oxidative stress, neuronal death, inflammation onset and tau phosphorylation/beta deposition. This leads to pathological changes in traumatic brain injury and ultimately increases the risk of neurological decline and neurodegenerative disease.

**Table 1 ijms-23-13850-t001:** List of the published clinical research of the drugs for copper.

NO.	Clinical Trial Identifier	Condition/Disease	Condition/Disease	Number of Participants
List of the published clinical trials that highlight the application of copper chelators				
1	NCT04737278	Neuralgia Myalgia	Drug: Cunermuspir Other: Placebo	56
2	NCT04422431	Wilson’s Disease	Drug: Bis-choline tetrathiomolybdate	31
3	NCT03539952	Wilson’s Disease	Drug: TETA 4HCL Drug: Penicillamine	53
4	NCT03299829	Trientine Treatment for Wilson’s Disease	Drug: Trientine	48
5	NCT02273596	Wilson’s Disease	Drug: ALXN1840	29
6	NCT01472874	Wilson’s Disease	Drug: Once a day trientine	8
7	NCT00325572	Autism Pervasive Developmental Disorder	Drug: Oral zinc and vitamin C supplements Other: Oral placebo	89
8	NCT00113542	Psoriasis	Drug: Tetrathiomolybdate (TM)	10
9	NCT00003751	Brain and Central Nervous System Tumors	Drug: Penicillamine Radiation: Radiation therapy	40
List of the published clinical trials that highlight the application of copper supply agent				
10	NCT03283800	Lipodermatosclerosis Chronic Venous Insufficiency Venous Insufficiency Varicose Veins	Other: Copper-impregnated compression stocking Other: Normal compression stocking	16
11	NCT03034135	Recurrent Glioblastoma	Drug: Disulfiram/copper Drug: Temozolomide (TMZ)	23
12	NCT01971112	Upper Respiratory Infections Lower Respiratory Tract Infections	Dietary Supplement: Multivitamins and minerals	320
13	NCT01177579	Copper Deficiency	Dietary Supplement: Copper gluconate	70
14	NCT00001262	Kinky Hair Syndrome	Drug: Copper histidine	60

Source: data retrieved from International Clinical Trials Registry Platform.

## Data Availability

Not applicable.

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
