# Peer review of "The Role of Copper Homeostasis in Brain Disease"

_ijms, 2022, doi:10.3390/ijms232213850_

Round 1
Reviewer 1 Report
This is a relevant topic for brain disease. The review is well structured.
Nevertheless, the manuscript requires English revision - as examples: pg.2 line 9 and pg.8 line 19.
The quality of the figures require improvement.
Author Response
Dear editor,
Thank you very much for your letter and the reviewers’ comments concerning our manuscript. Those comments are all valuable and very helpful for revising and improving our paper, as well as the important guiding significance to our researches. We mark all the changes in red in the revised manuscript.
Reviewer 1
This is a relevant topic for brain disease. The review is well structured.
- Nevertheless, the manuscript requires English revision - as examples: pg.2 line 9 and pg.8 line 19.
Response (R): According to the reviewer's good instruction, we have revised them.
Excess or deficiency of copper can lead to serious illness or death. Because copper has a powerful redox capacity, which can lead to oxidative stress and neurodegeneration within the brain.
AβPP is important for regulating Aβ production, while Aβ aggregates can produce ROS in the presence of copper ions and excess ROS is harmful to the brain.
2.The quality of the figures require improvement.
Response (R): According to the reviewer's good instruction, we have revised it.
Reviewer 2 Report
The manuscript "The role of copper homeostasis in brain disease" is a relevant review to update the information about Copper and the Brain in normal and pathological conditions and is appropriate for its publication. It is well-written and the figures included are appropriate to illustrate the text; although the resolution could be improved.
I will recommend including more recent references to experimental articles given that references from 2018 to the present are the fewest in number. Also, references to specific mechanisms regarding the experimental evidence around copper homeostasis and neuronal activation and, in general, specific to neurons and glia are poorly described.
Author Response
Dear editor,
Thank you very much for your letter and the reviewers’ comments concerning our manuscript. Those comments are all valuable and very helpful for revising and improving our paper, as well as the important guiding significance to our researches. We mark all the changes in red in the revised manuscript.
Reviewer 2
The manuscript "The role of copper homeostasis in brain disease" is a relevant review to update the information about Copper and the Brain in normal and pathological conditions and is appropriate for its publication.
- It is well-written and the figures included are appropriate to illustrate the text; although the resolution could be improved.
Response (R): According to the reviewer's good instruction, we have revised it.
- I will recommend including more recent references to experimental articles given that references from 2018 to the present are the fewest in number. Also, references to specific mechanisms regarding the experimental evidence around copper homeostasis and neuronal activation and, in general, specific to neurons and glia are poorly described.
Response (R): According to the reviewer's good instruction, we have added more recent relevant references to this part.
Copper is essential for diverse neuronal functions. Copper induces microglia activation in substantia nigra pars compacta of C57BL/6J mice. In addition to this, copper activates BV2 cells and induces the release of inflammatory cytokines. In BV2 cells, copper induced oxidative stress and activated the NF-κB/P65 pathway, which interfered with mitochondrial autophagy and eventually led to BV2 cell death[1]. The most abundant glial cells in the central nervous system are astrocytes, which play important roles in health and disease. Under normal conditions, astrocytes are involved in important physiological processes, such as development and functional regulation of synapses and the blood-brain barrier, metabolic support of neurons, and production of neurotrophins[2,3]. Kardos et al. believe that cell-level copper signaling between neurons and astrocytes is also present and may play an important role in brain signal processing[4]. Considerable evidence has shown that memory deficits in rats with chronic copper poisoning are associated with copper deposition in the choroid plexus, astrocyte swelling, astrogliosis and neuronal degeneration in the cerebral cortex, and increased copper levels in the hippocampus[5]. Particularly, in some neuroinflammatory diseases, such as multiple sclerosis, the expression of copper transporters such as CTR1 on glial cells depends on TRKB, and TRKB has been shown to play a key role in neurotrophin-induced calcium flux production in glial cells and CTR1 upregulation in vitro. These process cause astrocytes to take up and release copper, which in turn leads to oligodendrocyte loss[6].
- Zhou, Q.; Zhang, Y.; Lu, L.; Zhang, H.; Zhao, C.; Pu, Y.; Yin, L. Copper induces microglia-mediated neuroinflammation through ROS/NF-κB pathway and mitophagy disorder. Food Chem Toxicol 2022, 168, 113369, doi:10.1016/j.fct.2022.113369.
- Sofroniew, M.V. Astrocyte Reactivity: Subtypes, States, and Functions in CNS Innate Immunity. Trends Immunol 2020, 41, 758-770, doi:10.1016/j.it.2020.07.004.
- Vainchtein, I.D.; Molofsky, A.V. Astrocytes and Microglia: In Sickness and in Health. Trends Neurosci 2020, 43, 144-154, doi:10.1016/j.tins.2020.01.003.
- Kardos, J.; Héja, L.; Simon, Á.; Jablonkai, I.; Kovács, R.; Jemnitz, K. Copper signalling: causes and consequences. Cell Commun Signal 2018, 16, 71, doi:10.1186/s12964-018-0277-3.
- Pal, A.; Badyal, R.K.; Vasishta, R.K.; Attri, S.V.; Thapa, B.R.; Prasad, R. Biochemical, histological, and memory impairment effects of chronic copper toxicity: a model for non-Wilsonian brain copper toxicosis in Wistar rat. Biol Trace Elem Res 2013, 153, 257-268, doi:10.1007/s12011-013-9665-0.
- Colombo, E.; Triolo, D.; Bassani, C.; Bedogni, F.; Di Dario, M.; Dina, G.; Fredrickx, E.; Fermo, I.; Martinelli, V.; Newcombe, J.; et al. Dysregulated copper transport in multiple sclerosis may cause demyelination via astrocytes. Proc Natl Acad Sci U S A 2021, 118, doi:10.1073/pnas.2025804118.